# Rethinking LLM Ensembling from the Perspective of Mixture Models

## Abstract

Model ensembling is a well-established technique for improving the performance of machine learning models. Conventionally, this involves averaging the output distributions of multiple models and selecting the most probable label. This idea has been naturally extended to large language models (LLMs), yielding improved performance but incurring substantial computational cost. This inefficiency stems from directly applying conventional ensemble implementation to LLMs, which require a separate forward pass for each model to explicitly compute the ensemble distribution. In this paper, we revisit this conventional assumption and find that ensembling in the context of LLMs is fundamentally different. Unlike conventional models, LLMs typically generate tokens by sampling from the output distribution rather than selecting the top prediction via argmax. This key distinction enables us to reinterpret LLM ensembling as a mixture model. Under this perspective, one can sample from the ensemble distribution by simply selecting a single model at random and sampling from its output, which avoids the need to compute the full ensemble distribution explicitly. We refer to this approach as the **Mixture-model-like Ensemble** (ME). ME is mathematically equivalent to sampling from the ensemble distribution, but **requires invoking only one model**, making it **1.78×-2.68×** faster than conventional ensemble. Furthermore, this perspective connects LLM ensembling and token-level routing methods, suggesting that LLM ensembling is a special case of routing methods. Our findings open new avenues for efficient LLM ensembling and motivate further exploration of token-level routing strategies for LLMs. Our code is available at https://anonymous.4open.science/r/Mixture-model-like-Ensemble/.

## 1 Introduction

> *"The work of science is to substitute facts for appearances, and demonstrations for impressions."*
>
> — John Ruskin

In both conventional machine learning and deep learning, model ensembling has been a well-established technique for improving performance by combining the outputs of multiple weaker base models (Opitz & Maclin, 1999; Polikar, 2006; Rokach, 2010). In classification tasks, a common practice is to average the predicted probability distributions from several models and select the class with the highest aggregated probability (Rokach, 2010). This idea has been naturally extended to large language models (LLMs). Specifically, when predicting the next token, researchers similarly average the output distributions of multiple LLMs and sample a token from the averaged distribution (Yu et al., 2024; Huang et al., 2024; Xu et al., 2024; Yao et al., 2024; Mavromatis et al., 2024), as illustrated in Figure 1(a). By leveraging the complementary strengths of individual models, this method can improve the quality of generated text.

However, in the context of LLMs, ensemble methods are often considered too expensive and inefficient for practical use. An ensemble of $n$ models requires $n$ forward passes, making inference $n$ times more expensive than with a single model. Although in theory, we can parallelize the ensemble by assigning each model to a separate device and executing forward passes simultaneously, this approach rarely achieves the expected speedup in practice. As shown in Table 4 and Figure 3, the parallel ensemble (CE (Parallel)) provides only a small speed improvement compared to the sequential ensemble (CE

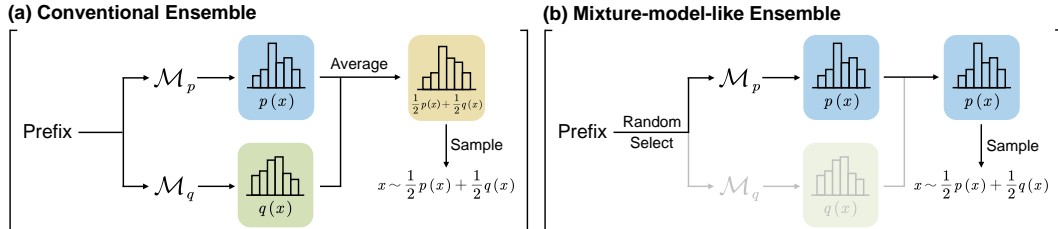

Figure 1: Comparison of (a) conventional ensemble and (b) mixture-model-like ensemble. $\mathcal{M}_p$ and $\mathcal{M}_q$ denote two distinct LLMs employed in the ensemble, with $p(x)$ and $q(x)$ indicating their respective output distributions.

(Sequential)), where each model is invoked sequentially. This is because parallel ensemble requires communication between devices for generating each token, and such overly frequent communication results in significant time overhead.

In addition, existing studies have attempted to mitigate the high inference overhead of LLM ensembles by reducing ensemble frequency (Yu et al., 2024) or limiting the vocabulary size (Yao et al., 2024). However, these methods offer only limited improvement. This is because, during ensemble, each model still must execute a forward pass to explicitly compute the ensemble distribution, which remains the core speed bottleneck unresolved. In this paper, we revisit this standard paradigm in LLM ensembling and pose a central question:

*Does a large language model ensemble truly require invoking all models?*

Surprisingly, we find that ensembling LLMs does not inherently require invoking all models. In fact, **ensembling $n$ LLMs can be achieved by invoking only one model**, resulting in inference speeds comparable to a single model. We hypothesize that the common belief that "LLM ensembling is slow" stems from the conventional machine learning ensemble paradigm, which has influenced both the perception and implementation of LLM ensembles. However, our findings reveal that LLM ensembling fundamentally differs from conventional ensemble methods.

Let's reconsider how conventional machine learning models make predictions. Typically, a model outputs a set of scores for each possible label, which are then normalized to sum to one using a function like softmax. These normalized scores are often *described as* a "probability distribution" over labels. However, they are not truly *used as* probabilistic distributions in practice. That is, we will not sample a label from this distribution as the final prediction. Instead, the standard practice is to select the label with the highest score as the prediction. This same approach is used when ensembling multiple models: We average the normalized scores from each model and select the label with the highest average score (Rokach, 2010). This requires running all base models and computing the averaged scores to make the final prediction.

In contrast, LLMs treat output distributions differently. During generation, tokens are typically sampled from the predicted distribution rather than selected via argmax. When ensembling LLMs, a common method is to sample from the average of the individual model distributions. Unlike previous methods, which require explicitly computing this ensemble distribution by invoking all models, we find that the equivalent result can be achieved by simply *select a single model at random and then sample from its distribution*. We prove that the resulting tokens have the same distribution as those sampled from the full ensemble, offering equivalent results with conventional ensembles and with significantly greater efficiency. This sampling procedure is analogous to sampling from a mixture model such as the Gaussian Mixture Model (Everitt & Hand, 1981), where one first selects a component at random and then samples from its distribution.

This mixture-model perspective leads to an equivalent but significantly faster ensemble algorithm, which we refer to as the Mixture-model-like Ensemble (ME). It also provides a conceptual bridge between LLM ensemble methods and token-level routing, where a router selects an appropriate LLM at each generation step to improve performance or efficiency (Belofsky, 2023; Muqeeth et al., 2024; Ostapenko et al., 2024). In token-level routing, if we randomly assign the input to one of the LLMs, the method becomes operationally equivalent to ME, where the output tokens follow the ensemble distribution. *From this perspective, LLM ensemble can be interpreted as the simplest case of routing.*

Token-level routing methods, by incorporating input-specific signals into the routing decision, may thus be viewed as a more general and potentially more effective extension of ensemble techniques.

We conducted a comprehensive evaluation of our mixture-model-like ensemble using diverse configurations, including ensembles of similar models, heterogeneous models, and models of varying sizes. We tested our method across four datasets and multiple GPU devices. The results demonstrate that its performance closely matches that of conventional ensembles, supporting our theoretical finding. Moreover, in terms of inference speed, the proposed method significantly outperforms both sequential and parallel conventional ensembles, achieving runtime efficiency comparable to that of a single model and approaching the theoretical limit. This significantly enhances the practicality of LLM ensembles in real-world applications. Our contributions are as follows:

1. We propose a new way of understanding LLM ensembling by framing it as a mixture model. To the best of our knowledge, this is the first work to formally establish this connection.

2. This perspective naturally lends to a new ensemble method, which we call the Mixture-model-like Ensemble (ME). ME is **1.78× - 2.68×** faster than conventional ensembling while achieving equivalent performance.

3. Furthermore, this perspective also provides a conceptual link between LLM ensembling and token-level routing. In particular, LLM ensembling can be interpreted as the simplest case of routing. This connection may offer valuable insights for future research.

## 2 RELATED WORK

**LLM ensembling**. LLM ensembling has been shown to enhance both performance and safety (Hoang et al., 2023; Li et al., 2024; Lu et al., 2024; Chen et al., 2025). Research on this topic primarily addresses two key challenges. First, methods are needed to align the vocabularies of different models. Approaches to this include using a unified vocabulary (e.g., a union of all vocabularies) (Yu et al., 2024; Yao et al., 2024; Phan et al., 2024) or transforming outputs into a shared, latent space (Huang et al., 2024; Xu et al., 2024). Second, researchers explore how to configure the ensemble, focusing on the choice of base models and their respective weights (Yao et al., 2024; Yu et al., 2024; Mavromatis et al., 2024).

Our work aims to provide an understanding of LLM ensembling from the perspective of mixture models. While issues like vocabulary alignment, model selection, and weight tuning are important, they are treated as orthogonal to our main focus.

**Token-level routing & Mixture-of-Experts (MoE).** Token-level routing methods typically involve training a router and dynamically routing the current input to the most appropriate model (Belofsky, 2023; Muqeeth et al., 2024; Ostapenko et al., 2024), while MoE involves training exports as well as the router, achieving better performance but involving more training parameters (Jiang et al., 2024; Dai et al., 2024).

ME, token-level routing, and MoE are all techniques that use efficient collaboration to improve LLMs. While they share this high-level goal, they differ in their fundamental approach and the specific trade-offs they make between training efficiency and performance.

As shown in Table 1, the primary distinction among these methods lies in their training and performance characteristics. In terms of inference, all three maintain a similar speed to that of a single model, making them considerably faster than conventional ensemble methods, which require executing $n$ separate models. However, their training requirements and corresponding performance gains vary greatly:

Table 1: Comparison of ME and other related works.

| Method | Inference Efficiency | Training Efficiency | Performance Improvement |
|---|---|---|---|
| CE | ★ | ★★★ | ★ |
| ME | ★★★ | ★★★ | ★ |
| Routing | ★★★ | ★★ | ★★ |
| MoE | ★★★ | ★ | ★★★ |

**ME** offers a simple, "plug-and-play" solution with no additional training cost, but it provides a relatively modest performance improvement. **Token-level routing** requires the extra cost of training a separate router but can yield greater performance gains. **MoE** represents the highest training

investment, requiring not only the training of a router but also the training of each expert's parameters from scratch. However, this higher cost typically translates into the most substantial performance boost.

Ultimately, MoE is the preferred choice when the primary objective is to maximize performance and computational resources are not a concern. In contrast, for a simple, low-cost approach to enhancing a model's performance without any additional training, ME stands out as a highly practical solution.

# 3 MIXTURE-MODEL-LIKE ENSEMBLE

## 3.1 CONVENTIONAL LLM ENSEMBLE

Unlike standard decoding, which relies on a single model's output distribution to generate the next token, conventional LLM ensembling combines outputs from multiple models during decoding. In particular, it aggregates the predicted next-token distributions from $n$ different LLMs through averaging or weighted averaging to form an ensemble distribution. The next token is then sampled from this combined distribution. Prior studies have shown that this approach often leads to better performance compared to using a single model alone.

Specifically, let the input prefix be $x_{\leq t}$, where $t \in \mathbb{N}$ represents its length. Let $\mathcal{V}$ be the set of all possible tokens. We consider an ensemble of $n$ different LLMs, denoted $\mathcal{M}_1, \ldots, \mathcal{M}_n$. For a given prefix $x_{\leq t}$, each model $\mathcal{M}_i$ outputs a prediction distribution over the next token $y \in \mathcal{V}$, expressed as $\mathcal{M}_i(y \mid x_{\leq t})$. The ensemble assigns a weight $\lambda_i \geq 0$ to each model, where the weights sum to one: $\sum_{i=1}^{n} \lambda_i = 1$. Under this standard ensembling framework, the probability of generating token $y$ at position $t + 1$, conditioned on the prefix $x_{\leq t}$, is computed as the weighted sum of the individual model predictions:

$$P(x_{t+1} = y \mid x_{\leq t}) = \sum_{i=1}^{n} \lambda_i \mathcal{M}_i(y \mid x_{\leq t}). \tag{1}$$

During inference, each model in the ensemble performs a forward pass—either sequentially or in parallel—to produce its prediction distribution over the next token. These distributions are then combined as formalized in Equation (1) to form the ensemble distribution. The next token is then sampled from this ensemble distribution. This process is applied iteratively at each generation step to produce the full output sequence. The overall procedure is outlined in pseudo-code in Algorithm 1.

| **Algorithm 1** Conventional LLM Ensemble. |
|---|
| **Ensure:** base models $\mathcal{M}_1, \ldots, \mathcal{M}_n$; ensemble weights $\lambda_1, \ldots, \lambda_n$ ; prefix sequence $x_{\leq t}$. |
| 1: $S \leftarrow x_{\leq t}$ |
| 2: **while** *not finish* **do** |
| 3:    **for** $i = 1 \rightarrow n$ **do** |
| 4:       $P_i \leftarrow \mathcal{M}_i(y \mid S)$ |
| 5:    **end for** |
| 6:    $\bar{P} \leftarrow \sum_{i=1}^{n} \lambda_i P_i$ |
| 7:    $x_{t+1} \leftarrow \bar{P}$    ▷ Sample from ensemble |
| 8:    $S \leftarrow S + x_{t+1}$ |
| 9: **end while** |
| 10: **return** $S$ |

| **Algorithm 2** Mixture-model-like Ensemble. |
|---|
| **Ensure:** base models $\mathcal{M}_1, \ldots, \mathcal{M}_n$; ensemble weights $\lambda_1, \ldots, \lambda_n$ ; prefix sequence $x_{\leq t}$. |
| 1: $S \leftarrow x_{\leq t}$ |
| 2: **while** *not finish* **do** |
| 3:    ▷ Randomly select an index based on the ensemble weights $\lambda_1, \ldots, \lambda_n$ |
| 4:    $i \leftarrow \text{RANDOMINDEX}(\lambda_1, \ldots, \lambda_n)$ |
| 5:    $P_i \leftarrow \mathcal{M}_i(y \mid S)$ |
| 6:    $x_{t+1} \leftarrow P_i$    ▷ Sample from $P_i$ |
| 7:    $S \leftarrow S + x_{t+1}$ |
| 8: **end while** |
| 9: **return** $S$ |

## 3.2 MIXTURE-MODEL-LIKE ENSEMBLE

In conventional LLM ensemble methods, each model independently performs a forward pass, and their output distributions are explicitly averaged to form an ensemble distribution. This combined distribution is then used to sample the next token. While this approach is straightforward, it requires evaluating all models, which can be computationally expensive.

However, as previously discussed in Section 1, we find that this explicit computation is not necessary. Given a set of ensemble weights $\lambda_1, \ldots, \lambda_n$ over $n$ models, we can instead sample an index $i$ from the multinomial distribution $\text{Multinomial}(\lambda_1, \ldots, \lambda_n)$ and use only the corresponding model $\mathcal{M}_i$ to

generate the next token. This stochastic selection ensures that the overall distribution of generated tokens still matches that of the full ensemble.

To further illustrate this equivalence, consider a prefix $x_{\leq t}$. The probability of generating a token $x'$ from this mixture-model-like ensemble is given by $P(x = x' \mid x_{\leq t})$. Then, we have:

$$
\begin{aligned}
P(x = x' \mid x_{\leq t}) &= \sum_{i=1}^{n} P(\text{model } \mathcal{M}_i \text{ is selected}) \cdot P(x' \text{ is generated by } \mathcal{M}_i) \\
&= \sum_{i=1}^{n} \lambda_i \mathcal{M}_i(y \mid x_{\leq t}).
\end{aligned}
\tag{2}
$$

As shown in Equation (1) and Equation (2), both the mixture-model-like ensemble and the conventional ensemble yield identical token distributions, thereby ensuring equivalent generation quality.

This method introduces randomness earlier in the generation process. In typical sampling, while the token selection is random, it is drawn from a fixed distribution, so the sampled token follows that specific distribution. However, a mixture model introduces an additional level of randomness by also randomizing the choice of distribution itself. As a result, the sampled tokens reflect the combined behavior of all distributions in the mixture, rather than any single one.

During inference, at each token generation step, a single model from the ensemble is sampled based on the ensemble weights $\lambda_1, \ldots, \lambda_n$ and used to perform a forward pass, producing a probability distribution over the next token. A token is then sampled from this distribution. This process is repeated iteratively at each step to generate the full output sequence. The complete procedure is presented in pseudo-code in Algorithm 2.

### 3.3 ENSEMBLING WITH HETEROGENEOUS VOCABULARIES

In this section, we introduce how to apply ME to model ensembles that use heterogeneous vocabularies. In practice, different models often have their own unique vocabularies, which makes it impossible to average their probability distributions directly. To address this, prior work has developed various vocabulary alignment methods (Yu et al., 2024; Yao et al., 2024; Phan et al., 2024). These methods typically involve creating a unified vocabulary and then mapping each model's probability distribution to it before combining them.

Specifically, let's consider $n$ LLMs, $\mathcal{M}_1, \ldots, \mathcal{M}_n$, with their respective vocabularies, $\mathcal{V}_1, \ldots, \mathcal{V}_n$. Their next-token prediction distributions are $P_1, \ldots, P_n$. We can define a transformation function, $\mathcal{F}_i : P_i \to \tilde{P}_i$, that maps the probability distribution $P_i$ from vocabulary $\mathcal{V}_i$ to a new distribution $\tilde{P}_i$ on the unified vocabulary $\mathcal{U}$. The ensemble distribution from Equation (2) can then be rewritten as:

$$
P(x = x' \mid x_{\leq t}) = \sum_{i=1}^{n} \lambda_i \mathcal{F}_i \left[ \mathcal{M}_i(y \mid x_{\leq t}) \right], \quad x' \in \mathcal{U}.
\tag{3}
$$

ME can seamlessly integrate any vocabulary alignment method, making it suitable for models with different vocabularies. Specifically, in each generation step, ME simply needs to randomly select a model (e.g., $\mathcal{M}_i$), perform a forward pass to get its probability distribution $P_i(y|x_{\leq t})$, and then apply transformation function $\mathcal{F}_i$ to obtain the new distribution $\tilde{P}_i = \mathcal{F}_i[P_i(y|x_{\leq t})]$. A new token is then sampled from $\tilde{P}_i$. This means that in Algorithm 2, we simply replace line 5's $\mathcal{M}_i(y \mid x_{\leq t})$ with $\mathcal{F}_i[\mathcal{M}_i(y \mid x_{\leq t})]$. In this paper, we use UniTe (Yao et al., 2024) for vocabulary alignment.

### 3.4 UNIFYING LLM ENSEMBLING AND TOKEN-LEVEL ROUTING

Token-level routing is another strategy for enabling collaboration among LLMs. It typically involves a trained router and multiple specialized, heterogeneous models. At each generation step, the router selects the most suitable model based on the current input prefix $x_{\leq t}$, and the chosen expert generates the next token. This approach aims to enhance overall performance or efficiency.

We find that ME provides a natural bridge between LLM ensembling and token-level routing. Specifically, if we design a simple router that randomly selects one expert at each generation step to process the input, the resulting behavior is effectively equivalent to ME. From this perspective,

this random routing mechanism essentially ensembles multiple experts. Conversely, we can interpret LLM ensembling as the most basic version of token-level routing.

This unified view allows us to compare the two approaches in terms of the tradeoff between performance and training cost. While both achieve similar inference speeds, token-level routing offers the potential for better performance by making informed routing decisions based on input content. However, this benefit comes at the cost of training a router, which adds computational overhead. In contrast, LLM ensembling requires no additional training—once multiple models are available, they can be used directly—making it a training-free, plug-and-play alternative.

## 4 EXPERIMENTS

### 4.1 EXPERIMENTAL SETUP

**Models for ensemble.** To thoroughly evaluate our approach, we test a variety of model combinations, categorized into three key scenarios:

1. Ensembling similar models: This category combines models that share the same architecture and vocabulary but were trained on different datasets. For our experiments, we use (Qwen-3B[1], Qwen-Math-1.5B) and (Openchat, Nous-Hermes) for a two-model ensemble, and use (OpenHermes, Openchat, and Nous-Hermes) for a three-model ensemble.

2. Ensembling heterogeneous models: In this scenario, we combine models with distinct architectures, vocabularies, and training data. For our experiments, we use (Openchat, Deepseek-7B, Mistral-7B).

3. Ensembling models of different sizes. For this scenario, we use (Llama-3-8B, Llama-3-1B) and (Llama-3-8B, Llama-3-3B).

This configuration covers several mainstream model series, such as Qwen and Llama, ensuring a broad and robust evaluation.

**Datasets and evaluation.** We evaluated our method across multiple tasks, including mathematical reasoning (GSM8K (Cobbe et al., 2021)), multi-task understanding (MMLU (Hendrycks et al., 2021)), complex logical reasoning (BBH (Suzgun et al., 2022)), and general knowledge (ARC (Clark et al., 2018)). For each dataset, we measured both accuracy and speed. For accuracy, we ran five separate experiments and reported the average score. For speed, we measured the number of tokens generated per second. All speed tests were performed on an H100 GPU unless specified otherwise.

**Compared methods.** In our experimental setup, we evaluated three main approaches: 1) Single Model — using a single model for direct inference without ensembling; 2) Conventional Ensemble (CE), as described in Section 3.1; and 3) Mixture-model-like Ensemble (ME), as described in Section 3.2. Inspired by UniTe (Yao et al., 2024), we employ the top-$k$ ensembling strategy to align vocabulary and enhance performance.

To further analyze inference speed, we divided CE into two variants: sequential CE, which invokes each model one after another, and parallel CE, which runs models concurrently on separate GPUs. For the parallel CE, we use the GaC (Yu et al., 2024) implementation. Note that the speed we reported may differ from those in prior work (Yu et al., 2024; Yao et al., 2024), as we enable key-value caching—a configuration more reflective of real-world applications. Further details of the experimental setup can be found in Appendix A.3.

### 4.2 ENSEMBLING ON SIMILAR AND HETEROGENEOUS MODELS

Table 2 and 3 show the performance of ensembling using similar and heterogeneous models, respectively, while Table 4 presents their corresponding inference speeds. Our results lead to three key observations:

First, our experiments confirm that **ME and CE have equivalent performance**. As shown in Tables 2 and 3, ME's performance is consistently on par with CE, whether the ensemble includes two or

---

[1]To ensure brevity, we use abbreviations for model names. For example, Qwen-3B refers to Qwen2.5-3B-Instruct. A comprehensive list of all models and their full names can be found in Table 5.

Table 2: Performance comparison of ME and other baselines on ensembling similar models. The numbers in parentheses ($+x$ / $-y$) indicate the performance gain or drop of the ensemble model compared to the best single model.

| Model | GSM8K | MMLU | BBH | ARC |
|---|---|---|---|---|
| ❶ Qwen-3B | 79.77 | 66.75 | 51.94 | 81.81 |
| ❷ Qwen-Math-1.5B | 79.39 | 39.54 | 39.75 | 46.23 |
| *Two model ensembling:* ❶ + ❷ | | | | |
| CE ($k = 5$) | 83.14 (+3.37) | 66.05 (−0.70) | 52.74 (+0.80) | 81.14 (−0.67) |
| ME ($k = 5$) | 82.97 (+3.20) | 65.61 (−1.14) | 53.04 (+1.10) | 81.12 (−0.69) |
| CE ($k = 10$) | 82.62 (+2.85) | 66.67 (−0.08) | 52.25 (+0.31) | 81.57 (−0.24) |
| ME ($k = 10$) | 82.83 (+3.06) | 67.90 (+1.15) | 52.51 (+0.57) | 81.10 (−0.71) |
| ❸ Openchat | 68.02 | 56.47 | 44.85 | 73.39 |
| ❹ Nous-Hermes | 67.11 | 58.37 | 46.72 | 73.02 |
| ❺ OpenHermes | 67.59 | 59.84 | 47.13 | 75.25 |
| *Two model ensembling:* ❸ + ❹ | | | | |
| CE ($k = 5$) | 69.34 (+1.32) | 60.60 (+2.23) | 48.12 (+1.40) | 78.84 (+5.45) |
| ME ($k = 5$) | 69.11 (+1.09) | 60.95 (+2.58) | 47.33 (+1.22) | 78.78 (+5.39) |
| CE ($k = 10$) | 68.19 (+0.17) | 60.28 (+1.91) | 47.82 (+1.10) | 78.70 (+5.31) |
| ME ($k = 10$) | 68.74 (+0.72) | 60.63 (+2.26) | 47.25 (+0.53) | 80.06 (+6.67) |
| *Three model ensembling:* ❸ + ❹ + ❺ | | | | |
| CE ($k = 5$) | 69.05 (+1.03) | 60.60 (+0.76) | 47.82 (+0.69) | 78.38 (+3.13) |
| ME ($k = 5$) | 69.42 (+1.40) | 59.97 (+0.13) | 48.04 (+0.91) | 78.42 (+3.17) |
| CE ($k = 10$) | 68.47 (+0.45) | 61.19 (+1.35) | 46.87 (−0.26) | 77.34 (+2.09) |
| ME ($k = 10$) | 67.93 (−0.09) | 60.80 (+0.96) | 47.40 (+0.27) | 76.29 (+1.04) |

more similar or heterogeneous models. This finding strongly supports our theoretical conclusion, outlined in Section 3.2, that these two ensembling methods are fundamentally equivalent.

Second, **ME is significantly faster**. Table 4 shows that ME's inference speed is consistently much higher than both Sequential CE and Parallel CE across all configurations. Impressively, ME's speed approaches the maximum theoretical limit—the speed of a single model—which highlights its efficiency. We also observed that Parallel CE provides only a slight speed increase over Sequential CE. This is likely due to the significant overhead from frequent GPU communication during parallel implementation.

Third, **adding more models doesn't always improve performance**. While model ensembling generally boosts performance, there isn't a guarantee of continuous gains by adding more models. For example, on all datasets except MMLU, where the ❸ + ❹ + ❺ configuration did not perform better than ❸ + ❹. This suggests that the best number of models for an ensemble should be carefully chosen based on the specific task and the models' characteristics.

### 4.3 ENSEMBLING MODELS OF DIFFERENT SIZES

As shown in Figure 2, ensembling models of different sizes with ME presents a trade-off between performance and speed. This is because larger models typically offer better performance but are slower, while smaller models are faster but less accurate. By combining them, ME balances these two factors. The hyperparameter $\lambda$ controls this balance: a higher value for $\lambda$ prioritizes performance over speed. If $\lambda$ is set to either 0 or 1, ME is equivalent to using a single model.

Table 3: Performance comparison of ME and other baselines on ensembling heterogeneous models.

| Model | GSM8K | MMLU | BBH | ARC |
|---|---|---|---|---|
| ❻ Openchat | 68.02 | 56.47 | 44.85 | 73.39 |
| ❼ Deepseek-7B | 53.63 | 46.10 | 36.02 | 56.41 |
| ❽ Mistral-7B | 46.90 | 56.22 | 41.25 | 68.75 |
| *Three heterogeneous model ensembling: ❻ + ❼ + ❽* | | | | |
| CE ($k = 5$) | 69.20 (+1.18) | 57.98 (+1.51) | 45.61 (+0.76) | 75.00 (+1.61) |
| ME ($k = 5$) | 69.86 (+1.84) | 57.79 (+1.32) | 45.76 (+0.91) | 74.19 (+0.80) |
| CE ($k = 10$) | 68.23 (+0.21) | 58.50 (+2.03) | 44.96 (+0.11) | 78.69 (+5.30) |
| ME ($k = 10$) | 67.81 (−0.21) | 58.46 (+1.99) | 45.38 (+0.53) | 78.58 (+5.19) |

Table 4: Speed comparison of ME and other baselines. The numbers in parentheses indicate the speedup relative to Sequential CE. Individual model speeds (in gray) are provided for reference, but aren't directly comparable to the ensemble methods.

| Method | ❶ + ❷ | ❸ + ❹ | ❸ + ❹ + ❺ | ❻ + ❼ + ❽ |
|---|---|---|---|---|
| Single Model | 63.68 (1.95×) | 54.83 (2.01×) | 54.71 (3.13×) | 54.42 (3.16×) |
| CE (Sequential) | 32.71 (1.00×) | 27.16 (1.00×) | 17.43 (1.00×) | 17.21 (1.00×) |
| CE (Parallel) | 34.58 (1.05×) | 34.24 (1.26×) | 31.74 (1.82×) | 31.55 (1.83×) |
| ME (Ours) | **58.25 (1.78×)** | **51.33 (1.89×)** | **46.22 (2.65×)** | **46.17 (2.68×)** |

It's important to note that this trade-off management is not the main purpose of ME; it's an incidental benefit of ensembling models of different sizes. For tasks where the goal is specifically to control these trade-offs, ME may not be as effective as specialized methods like token-level routing (Zheng et al., 2025).

### 4.4 FURTHER ANALYSIS

**Speedup on other common devices.** Our primary speed tests were conducted on H100 GPUs, but we also evaluated speed on other common devices, including the RTX 3090, V100, and A100. For these tests, we used three pairs of models with varying sizes: (Qwen-1.5B, Qwen-Math-1.5B), (Qwen-3B, Qwen-Coder-3B), and (Qwen-7B, Qwen-Math-7B).

As shown in Figure 3, the results are consistent with our primary findings in Section 4.2: ME is significantly faster than both sequential and parallel CE, and its speed is comparable to that of a single model inference. These findings demonstrate the robustness of the ME method across different hardware configurations.

An interesting finding on the RTX 3090 was that parallel CE was slower than sequential CE. This is likely due to the slower inter-GPU communication speed of the RTX 3090, which introduces significant overhead and negates the benefits of parallelization.

**Ablation on lambda.** We conducted an ablation study on $\lambda$ using two model combinations: ❶ + ❷ and ❸ + ❹. The results, shown in Figure 4, reveal two key findings: 1) The performance of ME consistently aligns with CE across different values of $\lambda$. This consistency further supports the equivalence of the two methods; 2) The ensemble effect's sensitivity to $\lambda$ depends on the performance difference between the individual models. When the models have similar performance, the ensemble's result is not significantly affected by $\lambda$. Conversely, when there's a larger performance gap between the two models, the ensemble effect changes significantly and follows a monotonic trend, as shown in Figure 4 (b).

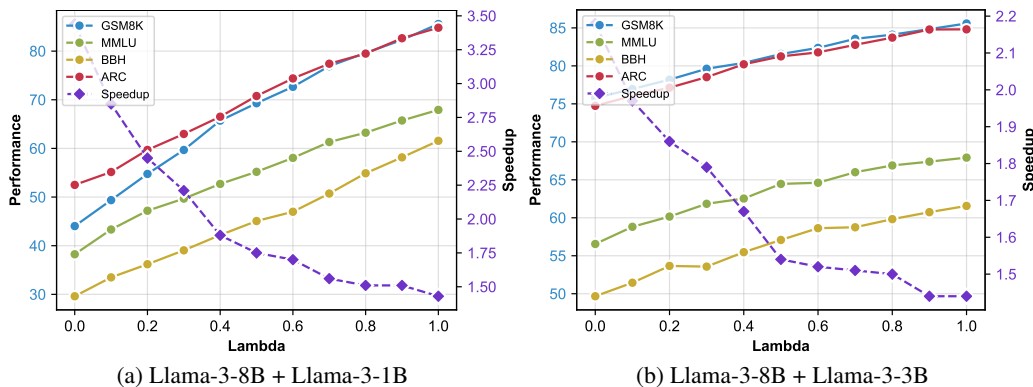

(a) Llama-3-8B + Llama-3-1B
(b) Llama-3-8B + Llama-3-3B

Figure 2: When ensembling models with different sizes, the trend of ME's performance and speed changing with $\lambda$. Here, $\lambda = 0$ indicates that only the smaller model is used for inference, while $\lambda = 1$ indicates that only the larger model is used.

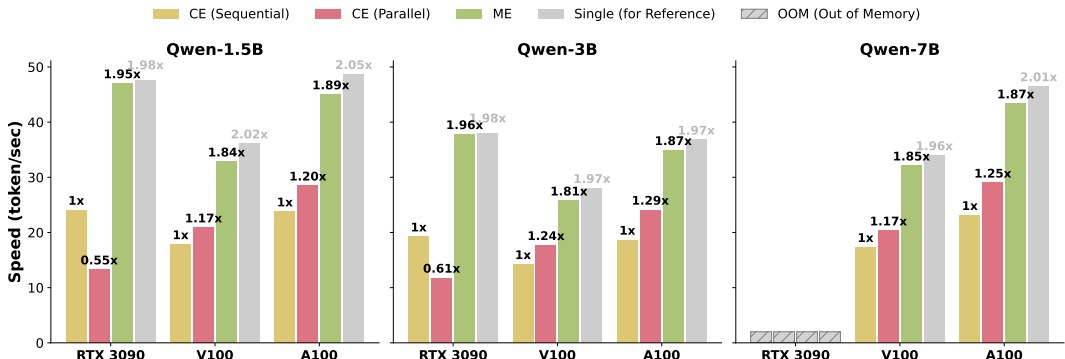

Figure 3: Speed comparison of ME and other baselines on three other common device types, using three model pairs of varying sizes.

## 5 DISCUSSION

**Limitations.** One limitation of our work is that the proposed mixture-model perspective is applicable when next-token sampling is used during generation—that is, when tokens are drawn probabilistically from the prediction distribution. However, in scenarios where greedy decoding is preferred—selecting the token with the highest probability—the mixture-model perspective no longer holds. In such cases, a full forward pass must still be performed for each model.

**Potential Extensions.** In this work, we focus on LLM ensemble methods that combine different model outputs using a weighted average. We believe this foundational approach can be extended to a wider range of combination strategies. A detailed discussion of these potential extensions is provided in Appendix A.5.

**Conclusion.** In this paper, we revisit the ensemble paradigm for large language models (LLMs) and introduce a novel perspective by framing LLM ensembling as a mixture model. From this viewpoint, we first naturally derive an algorithm that is equivalent in output to conventional ensemble methods but significantly more efficient. We term this algorithm the Mixture-model-like Ensemble. Second, we reveal a connection between two previously distinct research directions: LLM ensembling and token-level routing. We find that LLM ensembling can be interpreted as the most fundamental form of token-level routing. Extensive experiments across diverse datasets, model pairs, and GPU devices empirically support our findings. We hope this new perspective can provide valuable insights for future work on collaborative decoding in LLMs.

ETHICS STATEMENT

Our work adheres to the ICLR Code of Ethics. This study does not involve human subjects, personally identifiable information, or proprietary data. All datasets used in our experiments are publicly available. The proposed Mixture-model-like Ensemble (ME) method is a novel technique for improving the efficiency of large language model (LLM) ensembling. It does not introduce any new capabilities that could cause harm, nor does it enable misuse beyond the standard capabilities of existing LLMs.

We are not aware of any potential risks related to bias, fairness, or security that arise specifically from the proposed method. However, we acknowledge that the effectiveness and potential output of our method are dependent on the base LLMs used. As such, they may reflect or amplify biases present in the sources on which the base models were trained. While we did not perform a dedicated ethics audit, our approach does not introduce novel societal risks.

No conflicts of interest, legal compliance issues, or sponsorship-related influences are present in this work.

REPRODUCIBILITY STATEMENT

To ensure the reproducibility of our work, we have made several key efforts. Our proposed Mixture-model-like Ensemble (ME) method is mathematically simple and intuitive, with a clear and rigorous proof of its equivalence to conventional ensembling provided in Section 3.2 of the main text. Furthermore, the complete source code for our method and all experiments has been made publicly available in an anonymous repository at https://anonymous.4open.science/r/Mixture-model-like-Ensemble/. All datasets utilized in our experiments are publicly available, and we provide a comprehensive description of our experimental setup, hyperparameters, and data preprocessing steps in Section 4.1 and Appendix A.3 to enable full replication of our results.

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

## A APPENDIX

### A.1 THE USE OF LARGE LANGUAGE MODELS

This paper benefited from the use of Large Language Models (LLMs) as general-purpose assist tools. Specifically, LLMs were used for two main purposes:

- **Writing and Editing**: LLMs were used to refine and improve the clarity, grammar, and style of the manuscript.
- **Coding Assistance**: LLMs were utilized to assist with coding tasks, such as debugging and generating code snippets.

The authors take full responsibility for the content of this paper, including any parts that were generated or assisted by LLMs. LLMs were not involved in the research ideation, experimental design, or data analysis. They are not considered eligible for authorship.

### A.2 TESTED MODELS AND ABBREVIATIONS

Table 5: Tested Models and Abbreviations

| Model | Full Name |
|---|---|
| Qwen-3B | Qwen2.5-3B-Instruct (Yang et al., 2024a) |
| Qwen-Math-1.5B | Qwen2.5-Math-1.5B-Instruct (Yang et al., 2024b) |
| Openchat | Openchat-3.5-0106 (Wang et al., 2023) |
| Nous-Hermes | Nous-Hermes-2-Mistral-7B-DPO (Teknium et al., 2024) |
| OpenHermes | OpenHermes-2.5-Mistral-7B (Teknium, 2023) |
| Deepseek-7B | Deepseek-LLM-7b-Chat (DeepSeek-AI, 2024) |
| Mistral-7B | Mistral-7B-Instruct-v0.3 (Jiang et al., 2023) |
| Llama-3-8B | Llama-3.1-8B-Instruct (Dubey et al., 2024) |
| Llama-3-3B | Llama-3.2-3B-Instruct |
| Llama-3-1B | Llama-3.2-1B-Instruct |

### A.3 MORE EXPERIMENTAL DETAILS

For evaluation, we used the full test set of the GSM8K dataset. For MMLU and ARC, due to the size and category imbalance of their full test sets, we created balanced subsets. Specifically, we randomly selected 20 samples per subcategory from MMLU to form a 1,140-sample subset. For ARC, we sampled 50 instances from each subcategory to create a 1,350-sample subset.

All models were evaluated in a unified zero-shot setting with a temperature of 1. The specific prompts used are provided in our code repository. For our main results, we performed a grid search to find the optimal ensemble weights, using a step size of 0.1. The reported performance corresponds to this optimal weight configuration.

### A.4 ABLATION STUDY ON $\lambda$

### A.5 POTENTIAL EXTENSIONS

We observe that the mixture-model-like ensemble is applicable not only to LLM ensembles but also to broader forms of model combination. To illustrate this, consider a simple case where two probability distributions, $p(x)$ and $q(x)$, are combined into a new distribution $\mathcal{C}(p(x), q(x))$ through some combination operation. If this combined distribution partially contains the original distribution $p(x)$—specifically, there exists a parameter $\lambda \in (0, 1)$ such that the following inequality holds:

$$\mathcal{C}(p(x), q(x)) \geq \lambda p(x), \quad \forall x \qquad (4)$$

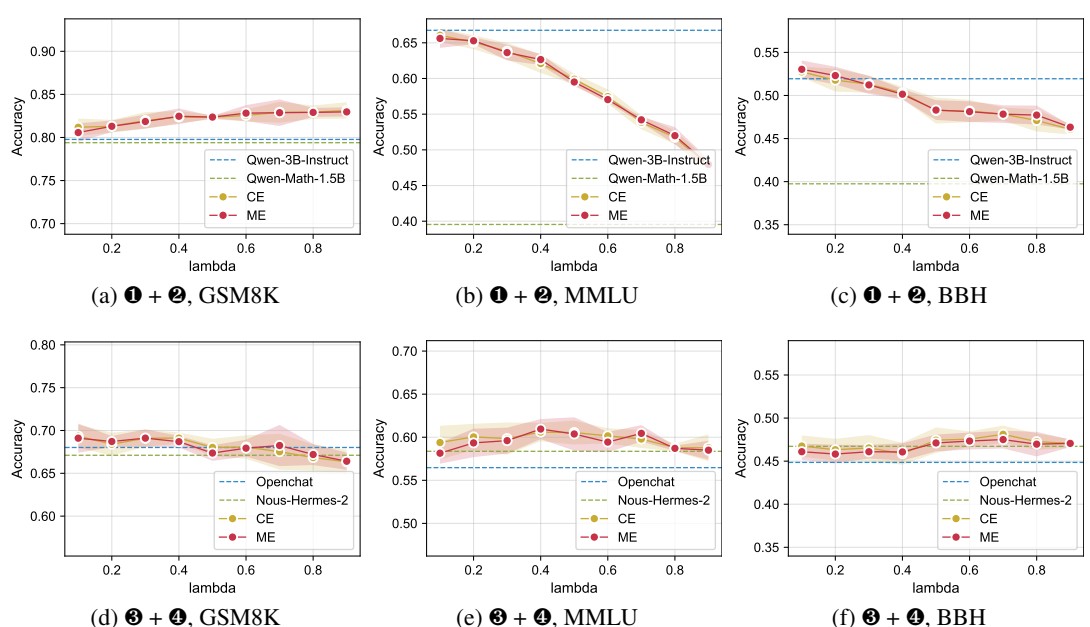

(a) ❶ + ❷, GSM8K  (b) ❶ + ❷, MMLU  (c) ❶ + ❷, BBH

(d) ❸ + ❹, GSM8K  (e) ❸ + ❹, MMLU  (f) ❸ + ❹, BBH

Figure 4: Ablation study on $\lambda$. $k$ is set to 5. Each point represents the mean of five independent runs, with the shaded bands showing the 95% confidence intervals.

Then, the combined distribution $\mathcal{C}(p(x), q(x))$ can be rewritten as:

$$
\begin{aligned}
\mathcal{C}(p(x), q(x)) &= \mathcal{C}(p(x), q(x)) - \lambda p(x) + \lambda p(x) \\
&= (1-\lambda)\left[\frac{1}{1-\lambda}(\mathcal{C}(p(x), q(x)) - \lambda p(x))\right] + \lambda p(x) \qquad (5) \\
&= (1-\lambda)\mathcal{C}'(p(x), q(x)) + \lambda p(x)
\end{aligned}
$$

where the new distribution is defined as:

$$
\mathcal{C}'(p(x), q(x)) = \frac{1}{1-\lambda}\left(\mathcal{C}(p(x), q(x)) - \lambda p(x)\right) \qquad (6)
$$

Clearly, by definition, $\mathcal{C}'(p(x), q(x))$ is non-negative and sums to 1; thus, it is a valid probability distribution.

Hence, we transform the original combined distribution $\mathcal{C}(p(x), q(x))$ into a mixture form of two distributions ($\mathcal{C}'(p(x), q(x))$ and $p(x)$), allowing us to apply the mixture-model-like ensemble for more efficient inference. Specifically, at each generation step, we sample from $p(x)$ with probability $\lambda$ and from the new distribution $\mathcal{C}'(p(x), q(x))$ with probability $(1-\lambda)$. Compared to the conventional combination method (which requires a forward pass from both models at each step), this approach only needs one forward pass under the $\lambda$ case, thus significantly improving inference efficiency.

This basic example can be extended to more complex formulations. For instance, the combination distribution $\mathcal{C}(p(x), q(x))$ may contains transformations of $p(x)$, such as $p(x)^2$ or Top-k$(p(x))$. In such cases, the generation process samples from $\mathrm{norm}\left(p(x)^2\right)$ or $\mathrm{norm}\left(\mathrm{Top\text{-}k}(p(x))\right)$ with probability $\lambda$.

This concept also generalizes to more model combinations. For example, given $n$ models in a combination distribution $\mathcal{C}(p_1(x), \ldots, p_n(x))$, one might form a sub-combination $\mathcal{C}'(p_{i_1}(x), \ldots, p_{i_k}(x))$ using only $k$ of the models ($k < n$). With probability $\lambda$, sampling is then restricted to these $k$ models, reducing computational cost and improving inference efficiency.

However, a systematic analysis of such extensions still requires further study. For instance, which combination structures are "separable," and how should the optimal separation strategy be determined? These issues are left for future work.

