# OpenReview forum: "Rethinking LLM Ensembling from the Perspective of Mixture Models"
_ICLR.cc/2026/Conference — ICLR 2026 Conference Withdrawn Submission_

### Official Review · Reviewer_2mRJ · 2025-10-14

**Soundness:** 4
**Presentation:** 4
**Contribution:** 2
**Rating:** 4
**Confidence:** 3

**Summary:**

The paper proposes a way to efficiently implement LLM ensemble. The method is based on a very simple math observation: Sampling from the mixture distribution can be implemented through ancestral sampling, i.e. first sampling a mixture index, then sampling from the mixture. In the setting of LLM ensembling, this corresponds to first sampling a LLM, then sampling a token from the picked LLM.

The paper then considers evaluation on different types of ensembling e.g. LLMs of similiar structure/vocabulary, or not. on standard benchmark tasks. The proposed ensemble method shows performance similiar to standard ensemble method (Table 2) but shows efficiency improvment (Table 4)

Overall I find the method technically sound, the presentation very clear, the paper is nicely written. However I have a couple of concern about the claim and contribution, see weakness and question section.

**Strengths:**

- The idea is simple and technically sound.

- The paper is well written and the presentation is very clear!

- The problem studied is of great importance.

- The experiments consider a wide range of models and tasks.

**Weaknesses:**

## Correctness of the claim

Does the equivalence between Alg.1 and Alg.2 still hold when top-P / top-K / temperature is used?

I think temperature would not affect the equivalence much but the usage of top-P / K would break the equivalence since the top-P / K of the weighted avearge distribution is different from that of the single model's distribution.

## Actual runtime and memory overhead

There are several things unclear about the efficiency gain of the proposed method compared with CE (Parallel).

- The proposed method would have the GPU-CPU load / unload overhead, which CE (Parallel) does not have.

- It is unclear how the proposed method manages KV cache: The KV cache needs to be loaded / unloaded from the memory to GPU everytime a model is picked; Many tokens' KV cache is not there for certain models when the models are not selected, so the corresponding kv caceh needs to be generated freshly.

- The CE(Parallel)'s actualy communication cost is unclear, and how exactly does the inter-GPU communication decompose is also unclear. Does the overhead mainly come from the exchange of next token prediction probability vector or does it come from the synchronization of different models on different machines due to their different sizes?

## Contribution

It also seems to me that many parts of the papers are not talking about the contribution of the proposed method in this paper, for example, the ablation study in Fig.2 can be applied to both CE and ME, and it is nothing particular about the proposed method.

**Questions:**

Is the efficiency difference between CE v.s. ME (parallel) mainly from
- At each token, CE (parallel)'s time is dominated by the slowest model, since each sampling step needs to wait for each model to finish the compute before aggregation.
- ME's time is roughtly the average of all models.
Therefore ME would be faster than CE. And the speed up would be more significant if the model sizes different are larger for all ensemble components?

---

> ### Author Response · Authors · 2025-12-03
> **Response to Reviewer 2mRJ**
>
> We sincerely thank the reviewer for their time and effort spent reviewing our submission and greatly appreciate your insightful comments and constructive suggestions. Below, we have done our best to address each of your concerns in detail.
>
> > **Weakness 1:** *Correctness of the claim*
>
> When using temperature, this equivalence still holds. As for top-P and top-K sampling, the prerequisite for their equivalence to hold is that: the top-P or top-K operation is first performed on each distribution separately, and then the ensemble is performed. For example, the UniTE model adopts this approach.
>
> > **Weakness 2.1:** *The proposed method would have the GPU-CPU load / unload overhead, which CE (Parallel) does not have.*
>
> In fact, ME does not incur GPU-CPU load/unload overhead. This is because when a model is not selected for execution, it is not unloaded, and thus does not cause additional load/unload overhead.
>
> > **Weakness 2.2:** *It is unclear how the proposed method manages KV cache: The KV cache needs to be loaded / unloaded from the memory to GPU everytime a model is picked; Many tokens' KV cache is not there for certain models when the models are not selected, so the corresponding kv caceh needs to be generated freshly.*
>
> Firstly, as we discussed above, ME does not unload the unselected models. Therefore, this avoids the additional  kv cache load/unload overhead.
>
> Secondly, when a model is selected for execution, it computes multiple kv caches at once that were not computed in previous steps. This process is similar to the prefill / extend operation in LLMs. You can refer to our code for detailed implementation specifics.
>
> > **Weakness 2.3:** *The CE(Parallel)'s actualy communication cost is unclear, and how exactly does the inter-GPU communication decompose is also unclear. Does the overhead mainly come from the exchange of next token prediction probability vector or does it come from the synchronization of different models on different machines due to their different sizes?*
>
> This extra overhead mainly stems from the exchange of the next token prediction probability vector. Since all the models are of comparable size in most of our tests, the synchronization time between different models is minimal.
>
> > **Weakness 3:** *It also seems to me that many parts of the papers are not talking about the contribution of the proposed method in this paper, for example, the ablation study in Fig.2 can be applied to both CE and ME, and it is nothing particular about the proposed method.*
>
> We apologize for any possible lack of clarity in our statements. We believe that the core contribution of this paper lies in the important fact that we proposed and proved: that "CE can be done by one forward pass".
>
> > **Question 1&2:**
>
> First, as we mentioned earlier, the inefficiency of CE (Parallel) does not primarily stem from waiting for each model to finish the compute, but rather from the overhead incurred by exchanging the next token prediction probability vector between different devices.
>
> However, CE (Parallel) also indeed needs to wait for each model to finish the compute. The performance improvement brought by ME becomes more significant when the size difference between models is larger. But this is mainly because CE wastes more time in this situation. For example, in Figure 2(a) (Llama-8B + 1B), ME can achieve a maximum speedup of 3.50x; whereas in Figure 2(b) (Llama-8B + 3B), ME can only achieve a maximum speedup of 2.2x.

---

### Official Review · Reviewer_XkeJ · 2025-10-17

**Soundness:** 3
**Presentation:** 4
**Contribution:** 2
**Rating:** 4
**Confidence:** 5

**Summary:**

This paper proposes a novel approach, termed Mixture-model-like Ensemble. It argues that conventional model ensemble approaches typically require forwarding n different models to generate one token, which results in significant computation overhead. The authors pointed out that with fixed ensemble weights, sampling from a weighted average of different models' output probabilities is mathematically equal to selecting one model and then sampling from its output probability. The new process would significantly reduce the computation overhead. Experiments across different models with diverse tasks demonstrate the effectiveness and efficiency of the proposed method.

**Strengths:**

1. The problem is important. Computation efficiency is indeed the main obstacle for the practical application of model ensemble approaches.
2. The idea is novel and effective. By converting sampling from a mixed distribution to first sampling the model, then generating a token, it largely reduces the need to forward n different models.

**Weaknesses:**

The main limitation of this work lies in its reliance on pre-defined ensemble weights. One of the most promising directions for token-level ensemble methods is the ability to dynamically adjust ensemble weights based on each model’s confidence at every step. Prior studies, such as EVA and SweetSpan, have shown that such adaptive weighting is key to achieving effective ensembling. Because a model’s performance can vary across different generation steps. An effective ensemble requires deciding the weights case by case, using the confidence or PPL as a proxy for model performance.

By contrast, the proposed approach fixes the ensemble weights in advance, which significantly constrains this potential and results in only marginal performance improvements. As shown in Appendix A.4, the optimal ensemble weights often converge toward 0 or 1, suggesting that the method may reduce to sampling from only the best-performing model in many cases.

Although the proposed approach does improve computational efficiency, it does so at the cost of limiting the performance gains that can be achieved through ensembling multiple models. The need to know the model's performance for deciding the ensemble weights in advance is also unrealistic in practice.

**Questions:**

1. What is the selected ensemble weight for the results reported?
2. What are the results for ensemble approaches that can dynamically adjust the ensemble weights?

---

> ### Author Response · Authors · 2025-12-03
> **Response to Reviewer XkeJ**
>
> We sincerely thank the reviewer for their time and effort spent reviewing our submission and greatly appreciate your insightful comments and constructive suggestions. Below, we have done our best to address each of your concerns in detail.
>
> In fact, the core focus of this paper is the equivalence between the ME distribution and the CE distribution. The applicability of ME is not limited only to the case of global weights; ME is equally applicable to the scenario of dynamic weights.

---

### Official Review · Reviewer_tjZs · 2025-10-28

**Soundness:** 3
**Presentation:** 2
**Contribution:** 2
**Rating:** 2
**Confidence:** 3

**Summary:**

The paper proposes an alternative to standard LLM ensembling: instead of running all $N$ models $f_1, \dots, f_N$ and averaging their next-token distributions, it samples one model at each step from a multinomial with weights $\boldsymbol{\alpha}$, and uses only that model’s output. The authors show this is faster than averaging distributions, with similar accuracy on their model pools as averaging.

My main concern is that it is underspecified. The method is presented as a training free approach compared to token routing, however it is a routing policy, just the input agnostic one. The paper calls this "training free" but does not explain how a user should actually choose these weights. Without a clear way to set , it is hard to evaluate how practical the method really is.

Moreover authors do not compare proposed forms of ensembling with prior ensembling works as well as token level routing (see Weaknesses). Overall, I have an impression that it is a token level routing method without a proper router but instead with predefined weights for each model. I think the approach would be much more convincing if the  coefficients were not fixed but learned as a function of model properties (e.g., size, performance, domain strength, latency, cost).

**Strengths:**

It is an interesting direction to reduce ensemble cost by sampling a single model instead of running all models at each step. The paper shows that this leads to speedups compared to averaging the output distributions.

**Weaknesses:**

1. Positioning

> While both achieve similar inference speeds, token-level routing offers the potential for better performance by making informed routing decisions based on input content. However, this benefit comes at the cost of training a router, which adds computational overhead. In contrast, LLM ensembling requires no additional training—once multiple models are available, they can be used directly—making it a training-free, plug-and-play alternative.

   This framing feels misleading. Training a lightweight router is typically negligible compared to training the expert models themselves, so it is unclear why router training is actually the bottleneck. If the authors believe it is, a citation would help. In addition, the proposed method is not really router-free: it still defines global mixture weights $\alpha_1,\dots,\alpha_n$ and samples an index from $\mathrm{Multinomial}(\alpha)$, which is itself a routing policy, just an input-agnostic one. The paper does not explain how a user is supposed to choose these $\alpha$ values in practice. Given that, it is not obvious why one should prefer fixed global weights over even a simple learned router that could account for task type, model size, latency/overhead constraints, and load balancing.

Authors do provide some ablations regarding the alpha parameter, but it is only for a pool of 2 models that were trained on the same data just different size (Llama 3b and 8b).

2. Lack of baseline comparison

As noted above, the paper positions the method relative to prior work, but the only quantitative baseline used is Conventional Ensembling (CE) -- plain averaging of next-token distributions. However, prior work on probability-level model ensembling (DeepEn, GAC, LLM-Blender, Pair-Ranker) applies additional post-processing, reweighting, or reranking over model outputs rather than naive averaging. These approaches are much closer to how ensembles are actually deployed and how the output distributions are used. The paper does not compare to these baseline. On the other hand, proposed method uses predefined input agnostic alphas (for me it is a form of routing) but do not compare with token level routing methods. Including such baselines, or explicitly explaining why they are not applicable here, would make the empirical claims significantly stronger.

Small notes:
1. I found the claim (“conventional machine learning models output scores, normalize with softmax, and then pick the highest-scoring label”) in the introduction vague and too broad. It is not clear what “conventional machine learning models” refers to. Are the authors specifically talking about multiclass classification? If so, this should be stated explicitly. More importantly, the contrast the authors set up between “softmax → argmax” and “sampling from a probability distribution” is not well motivated. Generative models (including language models, but also beyond LLMs) do treat the output as a probability distribution and sample from it. Conversely, taking argmax after softmax can be seen as sampling from a delta distribution. I would suggest tightening or removing this contrast, since it distracts from the core technical point.

**Questions:**

1. > An interesting finding on the RTX 3090 was that parallel CE was slower than sequential CE. This is likely due to the slower inter-GPU communication speed of the RTX 3090, which introduces significant overhead and negates the benefits of parallelization.

I suppose it depends on the type of interconnect that is used. Can authors clarify on that?

2. If I understand correctly, the proof of equivalence (Eq. (1) vs Eq. (2)) assumes you’re operating with the full distributions, but authors also use Top-k union (select the most probable k tokens). Then, if I understand correct, the equivalence breaks. Can authors clarify this?

3. What is $k$ that was used for the number in the Table 4?

4. Why in Table 2 the set of used models is different from Table 4? Could authors provide the downstream performance results for the models in Table 4? Also the models used in Table 4 are base model + math / code fine-tuned. I am not sure that this is the best use case for ensembling.

5. What alphas are used for the pool of 3 models in  Table 4?

---

> ### Author Response · Authors · 2025-12-03
> **Response to Reviewer tjZs**
>
> We sincerely thank the reviewer for their time and effort spent reviewing our submission and greatly appreciate your insightful comments and constructive suggestions. Below, we have done our best to address each of your concerns in detail.
>
> > **Weakness 1:** *Positioning*
>
> Firstly, training a router requires additional effort, including steps like data preparation and training. More importantly, a trained router has poor scalability. For example, when we change model sets, we need to re-train a new router. Therefore, we believe that training-free methods to achieve collaboration among models are valuable and meaningful.
>
> In fact, ME's focus is not on how to select $\alpha$, but is dedicated to proving that: "After selecting $\alpha$, applying ME can effectively achieve acceleration." Furthermore, ME is not limited to the scenario of global weights; ME is also applicable to scenarios with dynamic or learnable $\alpha$.
>
> > **Weakness 2:** *Lack of baseline comparison*
>
> As we discussed in the related work section, these methods are orthogonal to ours, and therefore, a direct performance comparison is unnecessary. This is because our core method (ME) can be combined with and accelerate many of these algorithms, such as GaC, Unite, and others.
>
> > **Question 1:** *I suppose it depends on the type of interconnect that is used. Can authors clarify on that?*
>
> Apologies, we may not have fully understood your question.
>
> Our understanding is: When applying parallel CE, a high-dimensional probability distribution vector needs to be transmitted between different devices. Because the bandwidth of the RTX 3090 is relatively low, transmitting this vector consumes more time, thus resulting in a large amount of extra overhead.
>
> > **Question 2:** *If I understand correctly, the proof of equivalence (Eq. (1) vs Eq. (2)) assumes you’re operating with the full distributions, but authors also use Top-k union (select the most probable k tokens). Then, if I understand correct, the equivalence breaks. Can authors clarify this?*
>
> This equivalence also holds when performing Top-P or Top-K operation on each distribution individually before performing the ensemble. For example, the UniTE model adopts this approach, and we also adopt UniTE's strategy in this paper.
>
> > **Question 3:** *What is $k$ that was used for the number in the Table 4?*
>
> During the benchmarking process, we uniformly used $k=5$. We believe that since the choice of $k$ has a negligible effect on the speed of ME and all baselines, we did not make a detailed distinction of the performance under different $k$ values here.
>
> > **Question 4:** *Why in Table 2 the set of used models is different from Table 4? Could authors provide the downstream performance results for the models in Table 4? Also the models used in Table 4 are base model + math / code fine-tuned. I am not sure that this is the best use case for ensembling.*
>
> Regarding Table 2 and Table 4, we adopted the same models.
>
> I speculate that your actual question is about the models used in Figure 3. In Figure 3, our core purpose is to demonstrate the performance of various methods in terms of inference speed in the ensembles scenario across different Model Sizes. Therefore, we chose the Qwen series model family, the reason being that they offer diversified model sizes.
>
> > **Question 5:** *What alphas are used for the pool of 3 models in Table 4?*
>
> In Table 4, we uniformly adopted the configuration of $(\frac{1}{3}, \frac{1}{3}, \frac{1}{3})$. We emphasize that, given similar model sizes, the speed of all methods is irrelevant to the chosen $\alpha$ value. Therefore, even if the configuration $(\frac{1}{3}, \frac{1}{3}, \frac{1}{3})$ is not optimal in terms of performance, it does not affect our results for speed testing and comparison.

---

### Official Review · Reviewer_NTcK · 2025-10-29

**Soundness:** 2
**Presentation:** 3
**Contribution:** 2
**Rating:** 2
**Confidence:** 4

**Summary:**

This paper revisits large language model ensembling from a theoretical and computational standpoint.
The authors argue that traditional ensemble methods—where multiple models’ output distributions are averaged before sampling—are unnecessarily expensive for LLMs.
They observe that because LLMs generate text via probabilistic sampling rather than argmax selection, the ensemble distribution can be equivalently realized by randomly choosing one model at each generation step and sampling from it.
Building on this observation, they propose a new algorithm, the Mixture-model-like Ensemble, which samples one model per token rather than performing multiple forward passes.
The paper proves that ME is mathematically equivalent to conventional ensembling, while being 1.78×–2.68× faster. Experiments on standard benchmarks (GSM8K, MMLU, BBH, ARC) demonstrate comparable performance between ME and traditional ensembles.
The authors further connect ME to token-level routing and mixture-of-experts methods, arguing that ensembling can be seen as a special case of routing.

**Strengths:**

1. The paper offers an reinterpretation of LLM ensembling as a mixture model, yielding a mathematically simple yet computationally efficient alternative to the standard approach. This theoretical reframing is insightful and connects two important areas—ensembling and token-level routing—in a unified probabilistic view.
2. The proposed Mixture-model-like Ensemble method is lightweight, requires no retraining, and provides substantial efficiency gains in inference, making it potentially appealing for applied settings where ensemble methods are otherwise too slow.
3. Experiments span diverse model families (Qwen, Llama, Mistral, etc.) and show consistent equivalence between ME and conventional ensembles across datasets, supporting the theoretical claim. The authors also evaluate speed on multiple GPUs, strengthening the empirical rigor.
4. The paper is well-structured and accessible. Figures and pseudo-code effectively illustrate the differences between conventional and mixture-model-like ensembles. The motivation is intuitive, and the empirical sections are easy to follow.

**Weaknesses:**

1. The equivalence between ME and conventional ensembling only holds when sampling is used for decoding. Most real-world LLM applications (QA, summarization, code generation) rely on deterministic decoding (greedy or beam search). The paper’s method thus applies to a narrower class of scenarios than implied, and no results are provided under deterministic conditions.
2. The theoretical proof guarantees equality of token-level distributions, but generation is autoregressive—errors or random choices can compound. The paper does not analyze how ME diverges from CE over long sequences or measure variance across multiple runs, leaving potential stability issues unaddressed.
3. While the authors draw conceptual parallels between ensembling and routing, no experiments compare ME with even simple token-level routing methods or mixture-of-experts variants. This weakens the claimed connection and leaves readers uncertain about ME’s relative merits beyond computational efficiency.
4. All benchmarks focus on factual or reasoning accuracy (e.g., GSM8K, MMLU). Since ensembles often facilitate diversity and alignment in generation, the omission of open-ended evaluations limits the significance of the findings.

**Questions:**

See weakness

---

> ### Author Response · Authors · 2025-12-03
> **Response to Reviewer NTcK**
>
> We sincerely thank the reviewer for their time and effort spent reviewing our submission and greatly appreciate your insightful comments and constructive suggestions. Below, we have done our best to address each of your concerns in detail.
>
> > **Weakness 1:** *The equivalence between ME and conventional ensembling only holds when sampling is used for decoding. Most real-world LLM applications (QA, summarization, code generation) rely on deterministic decoding (greedy or beam search). The paper’s method thus applies to a narrower class of scenarios than implied, and no results are provided under deterministic conditions.*
>
> As we mentioned in the discussion section (Section 5), we acknowledge that the proposed method is only applicable to sampling scenarios and not to deterministic decoding.
>
> However, we believe that sampling scenarios are not a "narrow" category. In fact, as the capabilities of LLMs themselves gradually saturate, researchers are increasingly focusing on utilizing sampling to build a larger search space to further enhance model performance. For instance, this includes applications such as Test-Time Scaling and Reinforcement Learning for LLM reasoning.
>
> > **Weakness 2:** *The theoretical proof guarantees equality of token-level distributions, but generation is autoregressive—errors or random choices can compound. The paper does not analyze how ME diverges from CE over long sequences or measure variance across multiple runs, leaving potential stability issues unaddressed.*
>
> As we demonstrated in the paper, ME is mathematically completely equivalent to CE. Therefore, the problem of error accumulation does not exist. It shares a core idea with Speculative Decoding, which is a lossless acceleration algorithm.
>
> > **Weakness 3:** *While the authors draw conceptual parallels between ensembling and routing, no experiments compare ME with even simple token-level routing methods or mixture-of-experts variants. This weakens the claimed connection and leaves readers uncertain about ME’s relative merits beyond computational efficiency.*
>
> As we stated in the related work section, the primary advantage of ME over token-level routing methods and MoE variants lies in its computational efficiency. Meanwhile, we also acknowledged in the related work section of our paper that ME is expected to achieve relatively lower performance, especially when compared to training-based methods. Therefore, we believe the conclusion is clear and explicit, and no additional comparison experiments are needed to further substantiate it.
>
> > **Weakness 4:** *All benchmarks focus on factual or reasoning accuracy (e.g., GSM8K, MMLU). Since ensembles often facilitate diversity and alignment in generation, the omission of open-ended evaluations limits the significance of the findings.*
>
> Thank you for your suggestion. We will consider adding the corresponding experiments in the subsequent version.

---

### Note · Authors · 2026-01-23

I have read and agree with the venue's withdrawal policy on behalf of myself and my co-authors.